# Identification of Oak-Barrel and Stainless Steel Tanks with Oak Chips Aged Wines in Ningxia Based on Three-Dimensional Fluorescence Spectroscopy Combined with Chemometrics

**DOI:** 10.3390/molecules28093688

**Published:** 2023-04-24

**Authors:** Yi Lv, Jia-Nan Wang, Yuan Jiang, Xue-Mei Ma, Feng-Lian Ma, Xing-Ling Ma, Yao Zhang, Li-Hua Tang, Wen-Xin Wang, Gui-Mei Ma, Yong-Jie Yu

**Affiliations:** 1Key Laboratory of Quality and Safety of Wolfberry and Wine for State Administration for Market Regulation, Ningxia Food Testing and Research Institute, Yinchuan 750004, China; 2College of Pharmacy, Ningxia Medical University, Yinchuan 750004, China; 3Key Laboratory of Ningxia Minority Medicine Modernization, Ministry of Education, Yinchuan 750004, China

**Keywords:** three-dimensional fluorescence spectroscopy, oak-barrel aged, wines, chemometrics

## Abstract

With the increased incidence of wine fraud, a fast and reliable method for wine certification has become a necessary prerequisite for the vigorous development of the global wine industry. In this study, a classification strategy based on three-dimensional fluorescence spectroscopy combined with chemometrics was proposed for oak-barrel and stainless steel tanks with oak chips aged wines. Principal component analysis (PCA), partial least squares analysis (PLS-DA), and Fisher discriminant analysis (FDA) were used to distinguish and evaluate the data matrix of the three-dimensional fluorescence spectra of wines. The results showed that FDA was superior to PCA and PLS-DA in classifying oak-barrel and stainless steel tanks with oak chips aged wines. As a general conclusion, three-dimensional fluorescence spectroscopy can provide valuable fingerprint information for the identification of oak-barrel and stainless steel tanks with oak chips aged wines, while the study will provide some theoretical references and standards for the quality control and quality assessment of oak-barrel aged wines.

## 1. Introduction

In recent years, wine has gained increasing popularity among Chinese consumers due to its antioxidant and antiseptic effects, which are considered to be beneficial to human health [1,2]. The organoleptic properties of wine depend to a large extent on the winemaking process and aging techniques used [3]. As is known to all, the winemaking process includes the harvesting of grapes, fermentation, aging, and bottling [4]. During the aging process, aging materials, baking degree (low, medium, high), and aging time are the important factors that affect the wine aging effect [5,6,7]. Generally speaking, the wine aging process requires careful consideration of the materials used, which is a key step in brewing high-quality wine [8]. The process of aging wines in oak barrels is a common practice that aims to improve their sensory quality and complexity [6]. On the one hand, the contact of the wine with the wood during fermentation and aging leads to a series of chemical reactions (condensation of anthocyanins and tannins, oxidation) that lead to significant changes in its chemical composition, which will eventually improve its performance [9]. On the other hand, the soluble compounds (phenolic compounds (PC), aromatic compounds (AC)) released from the wood into the wine contribute to its clarification and modify its astringency [10]. However, it is worth considering that wines can be aged in traditional ways during the winemaking process using alternative materials to oak barrels, such as steel and concrete, in order to obtain wines that are different from those already available on the market [11]. The development of these alternative materials is mainly due to the increasing global demand for wooden barrels (especially oak barrels) and the increasingly competitive wine market, with the aim of characterizing the chemical and physical characteristics of the wine during its aging in barrels [12]. Wine is usually aged in oak barrels for 6 to 18 months and the cost of the aging process is high, leading to a significant difference between the prices of commercially available aged wines in oak or stainless steel tanks with oak chips [13]. In this regard, the wine industry in China has witnessed a large number of fraudulent practices driven by economic interests, which have impacted both consumers’ interests and the development of the Chinese wine market. Regrettably, there are still no national standards or authoritative official methods to distinguish oak-barrel aged wines from stainless steel tanks with oak chips aged wines. Consequently, there is an urgent need to establish a reliable and fair method to identify oak-barrel and stainless steel tanks with oak chips aged wines, which is not only conducive to safeguarding the legal rights of consumers but also conducive to promoting the sustainable development of the wine industry and strengthening the crackdown by government supervision and management departments.

So far, most of the research on wines has focused on origin identification [14,15], quality control [16,17,18], grape varieties [19,20,21], grape years [22,23,24], and other aspects. The main analysis methods include gas chromatography–mass spectrometry (GC–MS), liquid chromatography–mass spectrometry (LC–MS), nuclear magnetic resonance technology (NMR), etc. Pan et al. [25]. established an untargeted metabolomic approach based on UPLC-QTOF-MS to discriminate the geographical origins of Chinese red wines. Wąsowicz et al. [26] used SPME-MS and SPME-GC/MS methods for the identification of white as well as red wines with the aim of distinguishing the varieties used for wine production and differentiating between wines based on the country of origin. Regarding the detection of metabolites in foods, nuclear magnetic resonance (NMR) was the most commonly used technique in the early days of metabolomics development. For example, Fan et al. [27] used ^1^H NMR spectroscopy combined with two chemometric methods, principal component analysis and linear discriminant analysis, to classify the varieties of Chinese red and white wine. However, the cumbersome and time-consuming sample pretreatment steps of the above technical means were likely to cause the loss of target detection objects and the consumption of a large amount of organic solvents. In addition, expensive instrument costs and maintenance costs were not beneficial to the application of these methods and the rapid discriminant analysis of wine samples. With regard to this, it is warranted to explore a new method that is free of chemical contamination, easy to operate, efficient in detection, and relatively low in cost.

In addition to the above studies, other researchers have also been committed to evaluating aged wines. Examples of these works include Anjos et al. [28] showed how to differentiate between aged wine spirits according to the aging technology (traditional, using 250 L wooden barrels, versus alternative, using micro-oxygenation and wood staves applied in 1000 L stainless steel tanks), the wood species used (chestnut and oak), and the aging time (6, 12, and 18 months) by FTIR-ATR. Apetrei et al. [29] conducted a study about the monitoring of evolution during red wine aging in oak barrels and an alternative method by means of an electronic panel test. Alañón et al. [30] investigated the volatile composition and sensorial characteristics of wines aged with chips and barrels of acacia wood with the aim of finding fingerprints able to discriminate between both types of aging. The main goal of the Gay et al. [31] article was to investigate the discrimination and classification of bottled wines previously aged in oak barrels from those previously treated with oak chips and oak staves by the electronic tongue. The majority of researchers have focused more on the changes in certain chemical components (anthocyanins [32], phenolics [33,34], polysaccharides [35], amino acid, organic acid [36], etc.) during wine aging. It is noteworthy that little research has been conducted to explore the classification potential between wines aged in oak barrels and stainless steel tanks with oak chips through the use of fluorescence spectroscopy. Consequently, it is necessary to establish a scientific and effective method to distinguish between oak-barrel and stainless steel tanks with oak chips aged wines.

Fluorescence spectroscopy, a new type of spectroscopic detection technology with rapid and high sensitivity, has developed rapidly in recent years with its unique advantages to facilitate the detection of food products with complex compositions. Especially, three-dimensional fluorescence spectrometry is preferred due to the fact that a large amount of fluorescence characteristics can be completely acquired by changing the excitation and emission wavelengths simultaneously [37]. The resulting emission–excitation data matrix (EEM) data consist of emission spectra registered at different excitation wavelengths. At the same time, owing to its characteristics of environmentally friendly, noninvasive, and nondestruction of samples, three-dimensional fluorescence spectroscopy combined with chemometrics has been widely used in the detection of various types of food such as grains and oil, meat, fruits and vegetables, wine, and other fields [38]. Wine is rich in phenolic acid, amino acid, epicatechin, tannin, anthocyanin, gallic acid, etc. All of these substances have fluorescent groups, which can emit fluorescence under the irradiation of a certain wavelength of light [39]. The fluorescence information contained in different excitation wavelengths reflects the total fluorescence information of the wine. The three-dimensional fluorescence spectral information of the wine serves as a stable system for the identification of the samples which can be regarded as a characteristic fingerprint. As a general conclusion, when combined with chemometric techniques, EEM facilitates the interpretation and extraction of information from the spectral data of wine samples, resulting in a powerful tool for wine characterization and classification.

Based on the above research status, this study used three-dimensional fluorescence spectroscopy combined with chemometric methods to analyze the three-dimensional fluorescence spectra of 87 wine samples, and used principal component analysis (PCA), partial least square discriminant analysis (PLS-DA), and Fisher discriminant analysis (FDA) to carry out pattern recognition research on wine fingerprints, which laid the foundation for the establishment of a low-cost, fast, and accurate method for the identification and detection of oak-barrel and stainless steel tanks with oak chips aged wines.

## 2. Results

### 2.1. Three-Dimensional Fluorescence Spectroscopic Analysis of Wine Samples with Different Dilution Factors

It is generally accepted that the concentration of fluorescent substances in fluorescence measurements should not be too high, otherwise fluorescence quenching will occur in direct measurements. In this study, the wine was dark, so the most suitable and reasonable dilution factor was first investigated by designing a series of dilution factors.

When the dilution factors were 5, 10, 20, 100, 500, 1000, 1500, and 2000, the three-dimensional fluorescence spectra of the wine quality control samples with different dilution factors were measured. The fluorescence peak locations (excitation wavelength/emission wavelength) and fluorescence intensity (I) of each wine sample are shown in Table 1. It can be seen from Table 1 and Figure 1 that there are obvious differences in the fluorescence peak intensities and positions of the wine samples with different dilution factors. The wine diluted with the 5 factor had the minimum fluorescence peak intensity at the excitation wavelength of 356 nm. The wine diluted with the 100 factor had the maximum fluorescence peak intensity at the excitation wavelength of 275 nm. What is clearly noticeable is the gradual decrease in the fluorescence peak intensity of the wine samples with the increasing dilution factors when the wine samples are diluted in the order of 100, 200, 500, 1000, 1500, and 2000 factors.

When the dilution multiples were 5, 10, 100, 500, 1000, 1500, and 2000, the contour maps of the wine quality control samples shown in Figure 2(A–C) represent the fluorescence peaks generated at different excitation wavelengths for different dilutions of the wine samples, and the intensity of the fluorescence peaks are reflected in Table 1. Although the wine samples exhibited very similar fluorescence properties, differences in the locations of the fluorescence peak, shapes, and the number of fluorescence peaks are easily seen in Figure 2. The locations and number of fluorescence peaks varied with the different dilution factors of the wine quality control samples. In Figure 2, the wine samples diluted with the 5 factor and 10 factor had different fluorescence peak locations and peak numbers. At a high concentration (Figure 2a, diluted with the 5 factor), the wine sample exhibited only one fluorescence peak (peak A) at an excitation wavelength of 356 nm. At a low concentration (Figure 2b, diluted with the 10 factor), three fluorescence peaks (peak A, peak B, peak C) appeared in the wine samples at excitation wavelengths of 338, 290, and 260 nm. At a low concentration (Figure 2c, diluted with the 20 factor), the wine samples showed three fluorescence peaks (peak A, peak B, peak C) at excitation wavelengths of 326, 287, and 260 nm. Since the fluorescence quenching phenomenon occurs at a high concentration, the fluorescence spectrum of the samples with the high concentration has fewer fluorescence peaks and lower fluorescence intensity than the low concentration. It can be seen from the information in Figure 2 that there were three fluorescence peaks in the wine samples diluted with the 10 factor and 20 factor, but the fluorescence peaks are comparatively obvious when the samples are diluted with the 10 factor. When the dilution factors of the wine samples were 100, 500, 1000, 1500, and 2000 (Figure 2d–h), the fluorescence peak positions were not significantly different and the number of fluorescence peaks was the same. The fluorescence quenching phenomenon was alleviated with the increase in the dilution factors. Therefore, taking into account the characteristics of the fluorescence intensities and the number of fluorescence peaks, the wine samples diluted with the 10 factor and 2000 factor were selected for the subsequent determination of the three-dimensional fluorescence spectra.

### 2.2. Analysis of Three-Dimensional Fluorescence Spectroscopy of Oak-Barrel and Stainless Steel Tanks with Oak Chips Aged Wines

According to analytical procedures, contour maps of 87 wine samples were obtained. Matlab software was used to draw the three-dimensional fluorescence spectra after deducting Rayleigh scattering and Raman scattering from the original measured spectral data. Rayleigh scattering and Raman scattering are manually deducted and the windows of both Rayleigh scattering and Raman scattering are ±12 nm.

The contour map of the wine samples before Rayleigh scattering and Raman scattering is shown in Figure 3. Figure 4 shows the contour maps of two representative wine samples diluted with the 10 and 2000 factor, containing the total fluorescence spectral information in the samples. A, B, and C represent the fluorescence peaks produced by each wine sample at different excitation wavelengths, respectively. As shown in Figure 4a, A is the fluorescence peak of the oak-barrel aged wines diluted with the 10 factor at the excitation wavelength/emission wavelength pair of 341/409, and its peak tip fluorescence intensity is 24.50; B is the fluorescence peak at the excitation wavelength/emission wavelength pair of 290/376, and its peak tip fluorescence intensity is 25.12; C is the fluorescence peak at the excitation wavelength/emission wavelength pair of 260/379, and its peak tip fluorescence intensity is 23.81. Studies suggest [21] that the peak A, the peak B, and the peak C could be (-)-Epicatechin, tryptophyl-lysine, and caffeic acid, respectively. Other graphical analyses are similar. In the analysis of other 3D fluorescence spectral patterns, the number, location, and fluorescence intensity of the fluorescence peaks generated by their samples can be visualized as well. It can be seen from Figure 4c that A was the fluorescence peak of the stainless steel tanks with oak chips aged wines diluted with the 10 factor at the excitation wavelength (nm)/emission wavelength (nm) of 338 nm/406 nm; B was the fluorescence peak at the excitation wavelength (nm)/emission wavelength (nm) of 290 nm/367 nm; C was the fluorescence peak at the excitation wavelength (nm)/emission wavelength (nm) of 260 nm/364 nm. While there were similarities in the number and location of the fluorescence peaks, there were differences in the intensity of the fluorescence peaks between the oak-barrel and stainless steel tanks with oak chips aged wines. This difference indicates that the fluorescent material may be related to the barrel fermentation process. In previous studies [40,41], anthocyanins and tannins in wines fermented in oak barrels decreased over time, and both components have fluorescent groups, presumably causing differences in intensity.

From the above analysis, it can be seen that although there are certain differences between the spectra of the oak-barrel and stainless steel tanks with oak chips aged wine samples, due to the similarity of fingerprint characteristics, the difference between the oak-barrel and stainless steel tanks with oak chips aged wines cannot be directly characterized by three-dimensional fluorescence spectroscopy. Therefore, the combination of excitation–emission fluorescence spectroscopy and chemometric methods are needed to achieve the purpose of differentiating between the oak-barrel and stainless steel tanks with oak chips aged wines

### 2.3. Classification and Discrimination of Oak-Barrel and Stainless Steel Tanks with Oak Chips Aged Wines

To further explore the potential of the data matrix, all the wine 3D fluorescence spectral data were expanded, transformed into 2D data, and normalized using Matlab, and then subjected to PCA, PLS-DA, and FDA to investigate the performance of multiple chemometric methods to differentiate between oak-barrel and stainless steel tanks with oak chips aged wines. The results of the cluster analysis for the oak-barrel and stainless steel tanks with oak chips aged wines are shown in Figure 5.

In order to observe the overall clustering effect and classification trend of the data matrix of wines, this study first used a widely used unsupervised pattern recognition method, PCA, to explore the differences between oak-barrel and stainless steel tanks with oak chips aged wines. The PCA model scoring plots for wines diluted with the 10 and 2000 factors are shown in Figure 5a and Figure 5b, respectively, and Figure 5c demonstrates the PCA model scoring plots combining the data from the diluted 10 and 2000 factors. The information obtained from Figure 5a,b indicates that the cumulative variance contribution of the first two principal components (PCs) to interpret wine EEM data diluted by 10 and 2000 factors accounts for 98.5% and 99.5%, respectively. In addition, the first two principal components accounted for 98.5 % when the wines were diluted with the 10 and 2000 factors (Figure 5c). Further observation of Figure 5 revealed that there was a large degree of overlap between the regions of the oak-barrel and stainless steel tanks with oak chips aged wine samples. The reason for this result may be the large variation in the samples within groups, which makes it difficult for the unsupervised analysis method to detect and distinguish the differences between groups. The above results suggested that the PCA model was unable to distinguish between the oak-barrel and stainless steel tanks with oak chips aged wines.

Given the above analysis, to address problems encountered in the unsupervised analysis and to assess the identification ability provided by the wine EEM data, a supervised pattern recognition method, PLS-DA, was employed to discriminate between samples from different groups. We employed the classic leave-one-out strategy to estimate LVs. To make the data analysis more efficient, a 10-fold was used, implying 90% of the samples were used for building the PLS-DA model and the remaining 10% were used for validation. The clustering results for the oak-barrel and stainless steel tanks with oak chips aged wines detected after the PLS-DA analysis based on the wine EEM data at dilutions of 10 and 2000 are presented in Figure 5d and Figure 5e, respectively, and Figure 5f demonstrates the results of the PLS-DA combining fusion data with dilution factors of 10 and 2000. It is noteworthy that the PLS-DA score plot showed that the samples of the stainless steel tanks with oak chips aged wines were relatively clustered (Figure 5d), while the samples of the oak-barrel aged wines were more scattered (Figure 5e). In addition, there were overlaps between the oak-barrel and stainless steel tanks with oak chips aged wine samples (Figure 5f), which may indicate that the distance between different wine sample points in the linear classification space is very close, making it difficult for PLS-DA, a method that specializes in solving linear classification problems, to meet the classification requirements. The above analysis results showed that the PLS-DA model could not completely discriminate the oak-barrel aged from the steel-barrel aged wines.

Fisher discriminant analysis (FDA) is a supervised classification that can reduce the distance between each sample point in the same class and expand the distance between each sample point in different categories, so as to effectively improve the discriminant efficiency. FDA has access to more comprehensive and richer information on wine characteristics, which is more conducive to classification. Based on these characteristic signals, we can observe the distribution of the characteristic data of the wine samples from a two-dimensional space and explore the influence of different features on the discriminatory ability by using the FDA discriminant model. In order to obtain satisfactory clustering results between oak-barrel aged wines and stainless steel tanks with oak chips aged wines, this study further constructed discriminant models based on the Fisher discriminant function for the oak-barrel and stainless steel tanks with oak chips aged wines, and performed multivariate discriminant analysis on the wine samples. The results of the FDA for the oak-barrel and stainless steel tanks with oak chips aged wines at dilutions of 10 and 2000 are presented in Figure 5g and Figure 5h, respectively, and Figure 5i shows the results of the FDA combining the fusion data with dilution factors of 10 and 2000. It can be clearly seen from the FDA that there is a satisfying clustering effect and classification trend between the oak-barrel and stainless steel tanks with oak chips aged wines. The intensification among the stainless steel tanks with oak chips aged wine samples was excellent (Figure 5i). The dispersion among the oak-barrel aged wines samples may be caused by two reasons (Figure 5g,h). One is that there is a certain difference in the fluorescence intensity of each oak-barrel aged wine; the other is that some features will be lost after the three-dimensional fluorescence spectral data of the oak-barrel aged wine are expanded and transformed into two-dimensional data, which leads to some bias in the analysis results. In addition, one of the stainless steel tanks with oak chips aged wine samples was incorrectly classified as an oak-barrel aged wine, probably because many unscrupulous vendors sold stainless steel tanks with oak chips aged wines as expensive oak-barrel aged wines in order to earn unjustified profits. The phenomenon of less overlap between individual oak-barrel and stainless steel tanks with oak chips aged wine samples may be due to the lack of explicit oak-barrel aged identification on commercially available wine bottles, resulting in confusion between oak-barrel and stainless steel tanks with oak chips aged wines. The FDA illustrates that three-dimensional fluorescence spectroscopy can distinguish between oak-barrel and stainless steel tanks with oak chips aged wine. The clustering effect shown in Figure 5 demonstrates that there was a better classification trend between the oak-barrel and stainless steel tanks with oak chips aged wines by using FDA than PCA and PLS-DA. Therefore, it is vital to establish a reliable and stable discriminant method for multivariate discriminant analysis of wine samples under different aging technology. In summary, the above analytical results indicate that EEM fluorescence spectroscopy combined with chemometric techniques has great potential in the classification of Chinese Ningxia wines.

## 3. Materials and Methods

### 3.1. Wine Samples

In this study, 87 wine samples were collected from the eastern foothills of Helan mountain, China, in 2022, including both oak-barrel aged (*n* = 36) and stainless steel tanks with oak chips aged (*n* = 51) wines. Wine samples were stored away from light in a refrigerator at −20 ℃ until further analysis. The distilled water was obtained from Watson’s Food & Beverage (Guangzhou, China). The pH was about 6.5~7.0. Detailed information of wine samples can be seen in Appendix A. A total of 87 wine samples were mixed in equal volumes for preparing wine quality control samples.

### 3.2. Three-Dimensional Fluorescence Spectroscopy Analytical Procedures

Due to the darker color of wine, the fluorescence quenching phenomenon would occur in direct measurement, thereby in the process of fluorescence analysis, serial dilution gradients were designed to determine the dilution factor according to the intensity of fluorescence peaks. The measurement time for a single sample can be as fast as 18 min. Through the results of the previous test, it was determined that the fluorescence peaks were relatively clear and obvious when the wine samples were diluted with the 10 factor and 2000 factor, so the samples in this experiment were all wines diluted with the 10 factor and 2000 factor with pure water that was purchased from Watsons Ltd. (Guangzhou, China).

Prior to fluorometric analysis, the selected wine samples were removed from storage and equilibrated to room temperature. The fluorescence spectra of 87 wine samples were directly measured on Hitachi F-7100 Fluorospectrophotometer (Hitachi, Japan) under the same conditions. Before the test, the configured wine samples were placed in the cuvette, the three-dimensional scanning method was selected, and the scanning parameters of the instrument were set. In this study, the excitation wavelength range of 200–500 nm and the emission wavelength range of 250–600 nm were chosen to be scanned at a scan rate of 2400 nm/min. The scan sampling was set to 3 nm, and both excitation and emission slits were set to 5 nm. In addition, the PMT voltage was 400 V. The spectra of the instrument components were calibrated using the manufacturer’s recommended calibration scheme (Hitachi F-7000 Owner’s Manual), which consists of both an excitation and emission calibration. After setup, the wine samples were scanned in 3D to obtain their detailed fluorescence spectral data.

### 3.3. Data Analysis

The three-dimensional fluorescence spectra of different wine samples were generated by Hitachi F-7100 Fluorospectrophotometer to obtain the 3D fluorescence spectral data of each sample. The EEM spectra demonstrate an elliptical contour. The x-axis represents the emission spectrum of 250–600 nm, and the y-axis indicates the excitation wavelength of 200–500 nm. The raw measured spectral data were processed using Matlab software and the 3D fluorescence spectrum was drawn after deducting Rayleigh and Raman scattering. The EEM spectra were vectorized and then auto-scaled (centered and normalized) before a chemometric analysis of the 3D fluorescence spectral data of the wine.

## 4. Conclusions

In this study, the 3D fluorescence spectra of 87 wine samples were analyzed using 3D fluorescence spectroscopy combined with chemometric methods. The Fisher discriminant analysis of the wine samples was able to obtain better clustering and differentiation, while the method was superior to PCA and PLS-DA in classifying oak-barrel and stainless steel tanks with oak chips aged wines. Overall, the simple and rapid wine quality identification method established in this study can provide a scientific basis for the national implementation of the protection of barrel-fermented quality wines, and also for the new technical method for the evaluation and analysis of the overall wine characteristics.

## Figures and Tables

**Figure 1 molecules-28-03688-f001:**
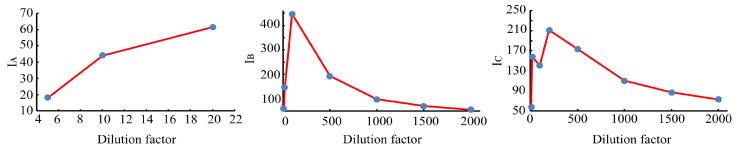
Fluorescence peak intensity of wine samples with different dilution factors.

**Figure 2 molecules-28-03688-f002:**
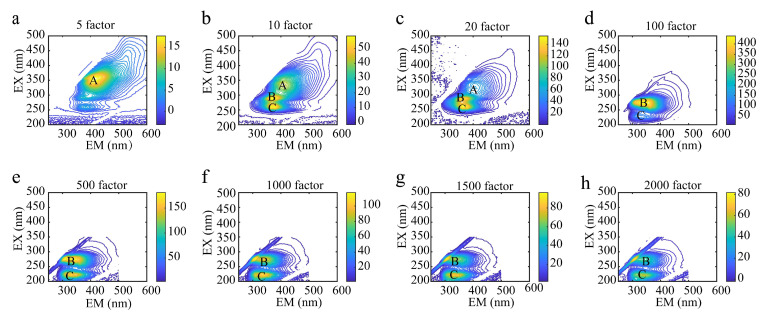
Contour maps of wines with different dilution factors. (A–C) represent the fluorescence peaks produced by each wine sample at different excitation wavelengths, respectively. (**a**–**h**) represent, respectively, the contour maps of oak-barrel and stainless steel tanks with oak chips aged wines diluted with different factors.

**Figure 3 molecules-28-03688-f003:**
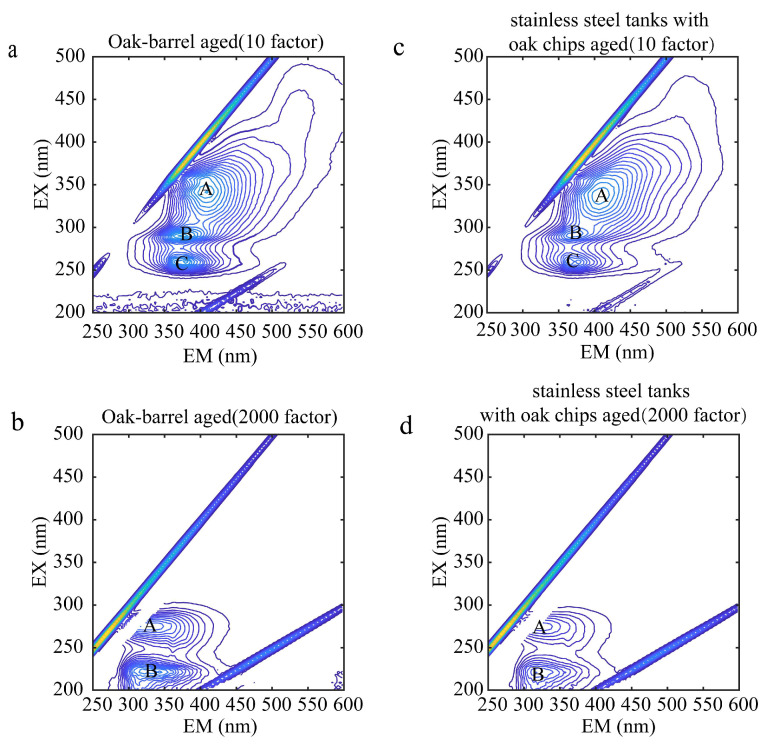
Contour maps of oak-barrel and stainless steel tanks with oak chips aged wines diluted with the 10 and 2000 factors before Rayleigh scattering and Raman scattering; (**a**,**b**) represent, respectively, the contour maps of oak-barrel aged wine samples diluted with the 10 and 2000 factors; (**c**,**d**) represent, respectively, the contour maps of stainless steel tanks with oak chips aged wine samples diluted with the 10 and 2000 factors. (A–C) represent the fluorescence peaks produced by each wine sample at different excitation wavelengths, respectively.

**Figure 4 molecules-28-03688-f004:**
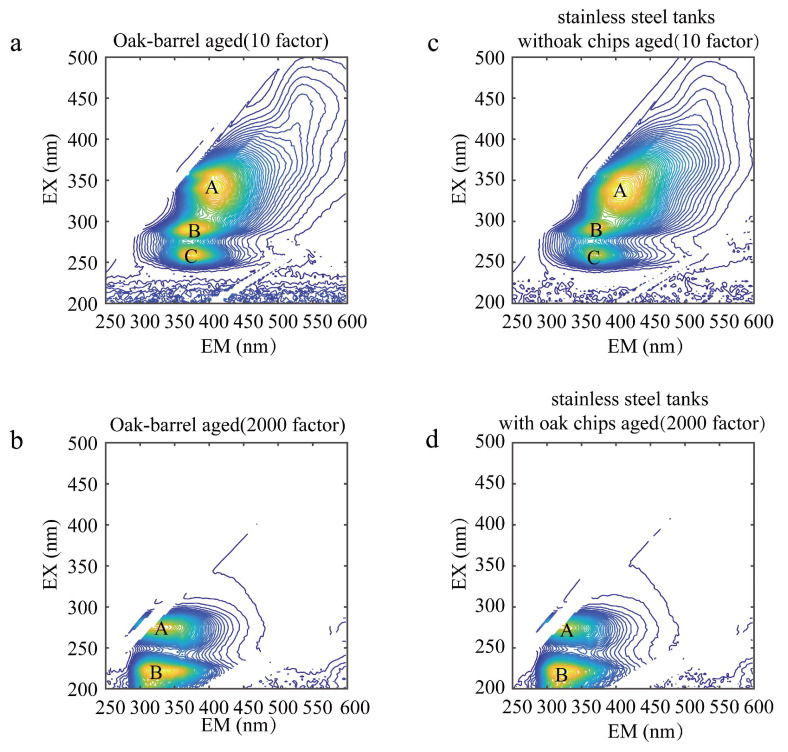
Contour maps of oak-barrel and stainless steel tanks with oak chips aged wines diluted with the 10 and 2000 factors; (**a**,**b**) represent, respectively, the contour maps of oak-barrel aged wine samples diluted with the 10 and 2000 factors; (**c**,**d**) represent, respectively, the contour maps of stainless steel tanks with oak chips aged wine samples diluted with the 10 and 2000 factors. (A–C) represent the fluorescence peaks produced by each wine sample at different excitation wavelengths, respectively.

**Figure 5 molecules-28-03688-f005:**
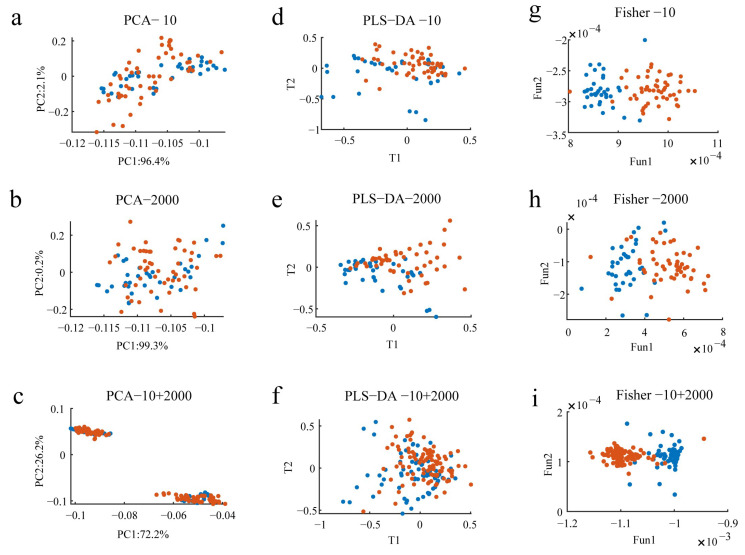
Scatter plot of multivariate analysis of wines. (**a**–**c**) represent the PCA model scoring plots for oak-barrel and stainless steel tanks with oak chips aged wine samples, respectively; (**d**–**f**) demonstrate the results of PLS-DA for oak-barrel and stainless steel tanks with oak chips aged wine samples, respectively; (**g**–**i**) show the results of FDA for oak-barrel and stainless steel tanks with oak chips aged wine samples, respectively. Blue dots represent wines aged in oak barrels, red dots show wines aged in stainless steel tanks with oak chips.

**Table 1 molecules-28-03688-t001:** Parameters of three-dimensional fluorescence characteristics of wines with different dilution factors.

Dilution Factor	Peaks Location λEX(nm)/λEM(nm)			Fluorescence Intensity(I)		
	A	B	C	I_A_	I_B_	I_C_
5	356/410	——	——	18.1	——	——
10	338/407	290/371	260/368	44.0	59.8	58.14
20	326/407	287/362	260/362	61.4	147.3	158.0
100	——	275/340	227/331	——	447.6	140.5
500	——	275/340	224/334	——	193.0	173.6
1000	——	275/343	221/340	——	97.5	109.9
1500	——	275/337	221/331	——	70.1	86.68
2000	——	275/340	221/337	——	55.9	72.78

## Data Availability

Not applicable.

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
