# Peer review of "Identification of Oak-Barrel and Stainless Steel Tanks with Oak Chips Aged Wines in Ningxia Based on Three-Dimensional Fluorescence Spectroscopy Combined with Chemometrics"

_molecules, 2023, doi:10.3390/molecules28093688_

Round 1
Reviewer 1 Report (Previous Reviewer 1)
See attached file

Author Response
Please see the attachment

Reviewer 2 Report (Previous Reviewer 2)
I still don't understand what a Steel barrel is? Is it a stainless steel tank without contact with the wood but with the size and shape of a barrel?
I still think that if the difference is without contact with wood in any format and in barrels, the difference is such that I don't see the justification for the work
In addition, the authors justify that the importance of their study is to differentiate wines with and without wood, but fraud exists in the different forms of wood, for this reason all the previous works have been carried out to differentiate wines with aging with alternative wood to the barrel and barrel aging. Not making sense another fraud. So the objective of the article I still do not understand. Please, if I am wrong, let me know, otherwise I could not accept this justification
I would need the authors to really explain to me why not do it like in the rest of the sites, and that this could be dealt with to avoid real fraud that is contact with wood in different ways, because otherwise there are much simpler analyzes for it, not seeing the interest in said work
Author Response
Please see the attachment

Reviewer 3 Report (New Reviewer)
The manuscript entitled “Identification of oak-barrel aged wines and steel-barrel aged wines in Ningxia based on three-dimensional fluorescence spectroscopy combined with chemometrics” explores the possibility to discriminate between wines aged with different oenological practices.
In my opinion the idea is great but the manuscript can not be published in this form and it needs major revision.
The English also needs a deep revision and in general the text must be checked for several mistakes and misprints.
The suggested revision as follow:
Title: please explain the meaning of steel -barrel. Usually it is not called steel barrel but wine aged in stainless steel tank. Please, clarify it through the manuscript as well.
Line 30-31: this sentence does not make sense because the winemaking does not include the planting
Line 34: I do not understand the meaning of “Due to wine”
Line 64, 71, 73: please delete the first name of the authors and leave just the last name.
Line 80-83: I thing that this is not true. In literature there are lots of papers about wine aging in oak and steel tanks in comparison for their chemical and sensory characterization during aging.
Line 95: do you mean gallic acid instead of gallnut acid?
Line 269-273: move this section to the section 3.2
Line 264-268: please explain what is the meaning of oak barrel aging and steel barrel aging. Please add information about the aging conditions (volume of the tanks, time of aging, temperatures….)
Figure 1. Move the description below the figure and explain better the figures i.e. Intensity (I)…
Figure 2: A B C are referred to which figures? Please add a better explanation.
Figure 4: please add the legend for the red and blue dots.
Author Response
Please see the attachment

Reviewer 4 Report (New Reviewer)
In the line 30 higlight that winemaking process and ageing techniques are responsible for organoleptic characteristics of wine (aroma and flavours) which are one of the most important factors in the selection of wine by consumers. Kindly consider to cite J. Serb. Chem. Soc. Vol. 88 No. 1 (2023) 11-23.
In the line 32 highlight that kinetic of phenolic compounds extraction during fermentation could also influence on the aging of wine. Kindly consider to cite Maced. J. Chem. Chem. Eng., 39, No. 2, (2020) 185–196.
Round 2
Reviewer 1 Report (Previous Reviewer 1)
Acceptable for publication
Reviewer 3 Report (New Reviewer)
The manuscript has been improved according the suggestions of the first revision. In particular, the specification of wine aged in stainless steel with oak chips changed completely the meaning of the research that at the beginning was very confusing.
Please, check again the text for some misprints such as i.e. Figure 1 caption shoul be put below the figure...
This manuscript is a resubmission of an earlier submission. The following is a list of the peer review reports and author responses from that submission.
Round 1
Reviewer 1 Report
Before deciding whether the manuscript can be accepted many changes must be made.
1) The wine samples are insufficiently described. The authors must indicate the criteria they used to determine whether the wines were oak-barrel aged or not.
No information is given concerning the grape varieties, the vintages (years of production), the wine producers, the regions, etrc.
Were ther any repetitions of the samples ? (not just of the instrumental measurements, but of the individual wines).
2) The authors don't indicate how the wines were diluted (probably with water, but at which pH ?)
3) The authors only show the scores for the first 2 PCs (for PCA) and LVs (for PLS-DA). There may be information on the other PCs and LVs.
4) They don't say whether the data was column-centered prior to analysis.
5) They do not say what method was used to normalize the data.
6) They do not discuss the origin of the two groups of samples in Figure 4C. (The two dilution levels ?)
7) LIne 209: They say they used PLS-DA to do a cluster analysis. THis is incorrect terminology. PLS-DA is a discriminat analysis method, not a clustering method.
8) They do not indicate how they determined the correct nuber of LVs for the PLS-DA. Did they do Cross-Validation ? Was there an external etst set to cvalidate their model ?
9) INsstead of plotting the PLS-DA scores, they could have plotted the predicted group values.
10) They do not propose any attributions for the peaks they observed in the EEM.
11) Lines 236-240: The authors indicate that they in fact do not know whether the wines are really oak-barrrel aged or not.
12) Line 260: What are "Watsons" ?
13) A reference should be given for AntDAS
14) How was the Rayleigh scattering and Raman scattering removed ?
15) No loadings are provided to help explain which fluorophores are responsible for the separation of the groups of samples.
16) In any case, applying PCA, PLS-DA and FDA to the unfolded EEMs is not the best way to analyse such 3-way data.
It would have been preferable to use Parafac or Independent Components Analysis.
17) From the description given, it is not clear whether they are "non-oaked barrel aged wines" or "non-aged wines".
In one case wines aged in steel for eaxmple, in the other case, wines that were not aged at all.
18) If the authors have access to a 3D Fluorescence spectrometer that can function in Front-Face mode, they could avoid the nedd to dilute the samples.
Reviewer 2 Report
COMMENTS
I believe that according to the work carried out and the information provided, it is not possible to discuss the work or draw conclusions.
INTRODUCTION
The first sentence of the introduction does not correspond, since the volumes of wine drunk have even decreased. If the authors want to keep it, they should modify it and put references
This sentence “According to the difference of fermentation 29 technology, wines sold in the market are divided into aged wines that have been fer-30 mented in oak barrels and non-aged wines that have not been fermented in oak barrels” it is very general, since there are many more techniques that differentiate wines, and if we talk about aging with wood, it could not only be divided into barrels and non-barrels. In aging with wood, we have barrel, aging with barrel alternatives (chips, staves...) and no contact with wood, and then within each of these modalities there are many factors that differentiate it.
Respect to this sentence, “Compared to non-oaked barrel aged wines, oak barrel aged wines can produce more complete fuller flavors, aromas, and softer textures, and their quality has also improved, leading to a significant difference in the price of commercially available wines”, is not only for wines aged in barrels but also for wines in contact with wood.
The problem of fraud is not the difference between wines aged in barrels and those not aged, because that is easy to detect, there are many compounds that differentiate them, the difficulty is differentiating wines aged in barrels and wines aged in wood in others formats. This is a problem that has existed in the world for many years and there are several methodologies that have given results, such as FTIR-ATR.
The introduction is not adequate, it talks about methods to classify by origins, varieties and does not talk about the methods that other authors have been studying to achieve the same objective that the authors had proposed. Therefore, the most important thing would be missing. Example of these works: Discrimination of aging wines with alternative oak products and micro-oxygenation by FTIR-ATR, Monitoring of evolution during red wine aging in oak barrels and alternative method by means of an electronic panel test, Fingerprints of acacia aging treatments by barrels or chips based on volatile profile, sensorial properties, and multivariate analysis, Application of an electronic tongue to study the effect of the use of pieces of wood and micro-oxygenation in the aging of red wine, among others
MATERIALS AND METHODS
There are many variables within aging with wood, barrel or alternative, MOX or no MOX, dose of wood, contact time, bottle time, wood species, toasting etc, so these factors of all the wines analyzed are not included. materials and methods and taking them into account at the time of the discussion could not lead to any conclusion
It is essential to indicate that it refers to non-oak barrel aged, because if it means without any type of contact with wood, there are a large number of compounds that differentiate them and therefore the work lacks interest since some are very quick to analyze. If the work has been carried out with different contacts with the wood, it is necessary to know what each of the 87 wines is
RESULTS AND DISCUSSION AND CONCLUSION
It would be necessary to know what has been done well, on what samples, to evaluate the differences well and to make an exhaustive study of it to contribute to the knowledge